# A Snapshot of European Children’s Eating Habits: Results from the Fourth Round of the WHO European Childhood Obesity Surveillance Initiative (COSI)

**DOI:** 10.3390/nu12082481

**Published:** 2020-08-17

**Authors:** Julianne Williams, Marta Buoncristiano, Paola Nardone, Ana Isabel Rito, Angela Spinelli, Tatjana Hejgaard, Lene Kierkegaard, Eha Nurk, Marie Kunešová, Sanja Musić Milanović, Marta García-Solano, Enrique Gutiérrez-González, Lacramioara Aurelia Brinduse, Alexandra Cucu, Anna Fijałkowska, Victoria Farrugia Sant’Angelo, Shynar Abdrakhmanova, Iveta Pudule, Vesselka Duleva, Nazan Yardim, Andrea Gualtieri, Mirjam Heinen, Silvia Bel-Serrat, Zhamyla Usupova, Valentina Peterkova, Lela Shengelia, Jolanda Hyska, Maya Tanrygulyyeva, Ausra Petrauskiene, Sanavbar Rakhmatullaeva, Enisa Kujundzic, Sergej M. Ostojic, Daniel Weghuber, Marina Melkumova, Igor Spiroski, Gregor Starc, Harry Rutter, Giulia Rathmes, Anne Charlotte Bunge, Ivo Rakovac, Khadichamo Boymatova, Martin Weber, João Breda

**Affiliations:** 1WHO European Office for the Prevention and Control of Noncommunicable Diseases, 125009 Moscow, Russian Federation; buoncristianom@who.int (M.B.); giuliarathmes@icloud.com (G.R.); anne-charlotte.bunge@charite.de (A.C.B.); rakovaci@who.int (I.R.); rodriguesdasilvabred@who.int (J.B.); 2Italian National Institute of Health (Istituto Superiore Di Sanità), 00161 Rome, Italy; paola.nardone@iss.it (P.N.); angela.spinelli@iss.it (A.S.); 3National Institute of Health Dr. Ricardo Jorge, 1600 560 Lisbon, Portugal; ana.rito@insa.min-saude.pt; 4Danish Health Authority, 2300 Copenhagen S, Denmark; thv@sst.dk; 5National Institute of Public Health, University of Southern Denmark, 1455 Copenhagen K, Denmark; leki@sdu.dk; 6Department of Nutrition Research, National Institute for Health Development, 11619 Tallinn, Estonia; eha.nurk@tai.ee; 7Institute of Endocrinology, Obesity Unit, 116 94 Prague, Czechia; mkunesova@endo.cz; 8Croatian Institute of Public Health, University of Zagreb, School of Medicine, 10000 Zagreb, Croatia; sanja.music@hzjz.hr; 9Spanish Agency for Food Safety and Nutrition, 28014 Madrid, Spain; mgarcias@mscbs.es (M.G.-S.); egutierrez@mscbs.es (E.G.-G.); 10Department of Public Health and Management, University of Medicine and Pharmacy Carol Davila, 030167 Bucharest, Romania; lbrinduse@gmail.com (L.A.B.); alexandra.cucu@insp.gov.ro (A.C.); 11Department of Cardiology, Institute of Mother and Child, 01-211 Warsaw, Poland; anna.fijalkowska@imid.med.pl; 12Primary Health Care, Ministry for Health, 1940 Floriana, Malta; victoria.farrugia-santangelo@gov.mt; 13National Center of Public health, Ministry of Health of the Republic of Kazakhstan, 010000 Nur-Sultan City, Kazakhstan; shynar_a@mail.ru; 14Centre for Disease Prevention and Control, LV-1005 Latvia, Riga; iveta.pudule@spkc.gov.lv; 15National Center of Public Health and Analyses, 1431 Sofia, Bulgaria; v.duleva@ncpha.government.bg; 16Turkish Ministry of Health, Public Health General Directorate, 34400 Istanbul, Turkey; nazan.yardim@saglik.gov.tr; 17Health Authority, Department of Health and Social Security, 47893 San Marino, San Marino; andrea.gualtieri.authority@pa.sm; 18National Nutrition Surveillance Centre, University College Dublin, Dublin, Ireland; mirjam.heinen@ucd.ie (M.H.); silvia.belserrat@ucd.ie (S.B.-S.); 19Republican Center for Health Promotion and Mass Communication, 720040 Bishkek, Kyrgyzstan; jama.usupova@mail.ru; 20Institute of Paediatric Endocrinology, National Medical Research Centre for Endocrinology of the Ministry of Health of the Russian Federation, 117036 Moscow, Russian Federation; peterkovava@hotmail.com; 21National Center for Disease Control and Public Health of Georgia, 0198 Tbilisi, Georgia; l.shengelia@ncdc.ge; 22Institute of Public Health, 1007 Tirana, Albania; lhyska2002@yahoo.it; 23Internal Diseases Department of the Scientific Clinical Centre of Mother and Child Health, 744036 Ashgabat, Turkmenistan; ovezmyradovag@who.int; 24Department of Preventive Medicine, Lithuanian University of Health Sciences, 44307 Kaunas, Lithuania; ausra.petrauskiene@lsmuni.lt; 25Department for Organization of Health Services to Children, Mothers, Adolescents and Family Planning, Ministry of Health and Social Protection of Population, 734025 Dushanbe, Tajikistan; sanavbar2010@list.ru; 26Institute of Public Health of Montenegro, 81 000 Podgorica, Montenegro; enisa.kujundzic@ijzcg.me; 27Applied Bioenergetics Lab, University of Novi Sad, 21000 Novi Sad, Serbia; sergej.ostojic@chess.edu.rs; 28Department of Pediatrics, Paracelsus Medical University, 5020 Salzburg, Austria; d.weghuber@salk.at; 29Arabkir Medical Centre-Institute of Child and Adolescent Health, 0014 Yerevan, Armenia; mmelkumova@mail.ru; 30Institute of Public Health, Faculty of Medicine, Ss. Cyril and Methodius University, 1000 Skopje, North Macedonia; i.spiroski@iph.mk; 31Faculty of Sport, University of Ljubljana, 1000 Ljubljana, Slovenia; gregor.starc@fsp.uni-lj.si; 32Department of Social and Policy Sciences, University of Bath, Bath BA2 7AY, UK; hr526@bath.ac.uk; 33WHO Country Office for Tajikistan, 734019 Dushanbe, Tajikistan; boymatovak@who.int; 34WHO Child and Adolescent Health and Development, WHO Regional Office for Europe, 2100 Copenhagen, Denmark; weberm@who.int

**Keywords:** nutrition, child, obesity, surveillance, health, noncommunicable diseases, children, fruit, vegetables, soft drinks

## Abstract

Consuming a healthy diet in childhood helps to protect against malnutrition and noncommunicable diseases (NCDs). This cross-sectional study described the diets of 132,489 children aged six to nine years from 23 countries participating in round four (2015–2017) of the WHO European Childhood Obesity Surveillance Initiative (COSI). Children’s parents or caregivers were asked to complete a questionnaire that contained indicators of energy-balance-related behaviors (including diet). For each country, we calculated the percentage of children who consumed breakfast, fruit, vegetables, sweet snacks or soft drinks “every day”, “most days (four to six days per week)”, “some days (one to three days per week)”, or “never or less than once a week”. We reported these results stratified by country, sex, and region. On a daily basis, most children (78.5%) consumed breakfast, fewer than half (42.5%) consumed fruit, fewer than a quarter (22.6%) consumed fresh vegetables, and around one in ten consumed sweet snacks or soft drinks (10.3% and 9.4%, respectively); however, there were large between-country differences. This paper highlights an urgent need to create healthier food and drink environments, reinforce health systems to promote healthy diets, and continue to support child nutrition and obesity surveillance.

## 1. Introduction

It is important for a child to eat a healthy diet in order to prevent malnutrition (stunting, wasting, micronutrient deficiencies, obesity) and noncommunicable diseases (NCDs) [1,2]. Low-quality diets are now believed to be the single biggest risk factor for the global burden of disease [3]. In recent decades, changes in dietary patterns and physical activity behaviors have been identified as likely contributors to a rise in childhood obesity [4,5]. Research from the latest round of the WHO European Childhood Obesity Surveillance Initiative (COSI) carried out in 2015–2017 indicates that 29% of boys and 27% of girls aged seven to nine years had overweight and there was a prevalence of obesity of 12% in boys and 9% in girls [6]. At the same time, in certain parts of the WHO European Region, there is a double burden of malnutrition, characterized by the coexistence of undernutrition (being underweight for one’s age, too short for one’s age (stunted), too thin for one’s height (wasted), or deficient in vitamins and minerals (micronutrient malnutrition)), along with overweight, obesity, or noncommunicable diseases, within individuals, households, and populations, and across the life course [7,8].

The prevalence of overweight and obesity is increasing worldwide [9]. According to the WHO Global Monitoring Framework for NCDs, which is a set of 25 indicators and 9 voluntary targets are used to track progress toward reaching global targets in 2015–2020, not a single country in the WHO European Region is likely to meet Global Monitoring Target 7, which aims to “halt the rise in diabetes and obesity” [10].

Ensuring that children consume healthy diets is important for achieving the UN Sustainable Development Goals (SDGs) related to no hunger (SDG Goal 2), good health and well-being (SDG Goal 3), quality education (SDG Goal 4), no poverty (SDG Goal 1), economic growth (SDG Goal 8), and more [11,12]. Food preferences and eating habits established in childhood and adolescence tend to be maintained into adulthood [13], making nutrition in childhood an important public health issue.

A healthy diet includes adequate quantities and appropriate proportions of fruit, vegetables, legumes (e.g., lentils and beans), nuts, and whole grains [14], and limits the intake of free sugars [15,16], salt [17], saturated fats, and highly processed foods. A healthy diet eliminates trans fats of all kinds. Consumption of sugar-sweetened beverages should be limited, as it has been associated with increased body weight [18] and dental caries [19].

The WHO European Childhood Obesity Surveillance Initiative provides data on the eating behaviors of children across the WHO European Region. Established in 2007, COSI collects high-quality data on the childhood obesity prevalence and energy-balance-related behaviors [20]. These data enable countries to set national targets, monitor trends over time, make comparisons between countries, and over time, to evaluate the effectiveness of obesity prevention efforts. In addition to collecting high-quality anthropometric measurements from primary-school-aged children, COSI also collects information on children’s dietary and physical activity patterns, screen time, sleep, and more.

Given the importance of nutrition in childhood, alongside the rising trend in childhood obesity, this study used the most recent results from the COSI study to describe the eating behaviors of children aged 6–9 years from across the WHO European Region.

## 2. Materials and Methods

Data were collected between 2015 and 2017, as part of COSI round four. Among the thirty-six countries participating in round four, 23 of them collected information on children’s dietary behaviors using parental reports on a “family form”. These countries were: Albania, Bulgaria, Croatia, Czechia, Denmark, Georgia, Ireland, Italy, Kazakhstan, Kyrgyzstan, Lithuania, Latvia, Malta, Montenegro, Poland, Portugal, Romania, Russian Federation (Moscow only), San Marino, Spain, Tajikistan, Turkey, and Turkmenistan. Children’s parents or caregivers were asked to complete a questionnaire that contained indicators of energy-balance-related behaviors (including diet) and family socioeconomic status. Completion of the form was voluntary and participants could opt out or choose not to participate at any time.

The COSI study follows the International Ethical Guidelines for Biomedical Research Involving Human Subjects [21]. Local ethics approval was also granted. Details for this approval are found in Appendix A. More details on the data collection procedures are provided elsewhere [20,22,23].

Parents were asked: “over a typical or usual week, how often does your child eat or drink the following kinds of foods or beverages”? This was followed by a tick box, where parents answered “never”, “less than once a week”, “some days (1–3 days)”, “most days (4–6 days)”, or “every day”. Parents were asked to report on a number of food items, shown in Appendix A. For this paper, we reported on the consumption of fresh fruits, vegetables (excluding potatoes), savory snacks (e.g., potato crisps, corn chips, popcorn, peanuts), sweet treats (e.g., candy bars or chocolates), and sugar-containing soft drinks. These questions were selected because they provided a summary that was related to common sources of nutrients of interest [24]. Countries chose country-specific examples for the food examples for “savory snacks (like potato crisps, corn chips, popcorn, peanuts)”, or “sweets (like candy bars or chocolate)”. These examples were identified by leading nutrition experts within the country and approved by the government-appointed principal investigator of the study. All questionnaires were translated from English into the local language, and then back-translated into English to check for discrepancies with the original English form.

For each country, we calculated the percentage of children who consumed these foods “every day”, “most days (four to six days per week)”, “some days (one to three days per week)”, or “never or less than once a week”. We reported these results stratified by country, sex, and region. Geographic regions were based on the United Nations Standard Geographical regions, which are based on continental regions and are further subdivided into sub-regions and intermediary regions that are drawn to obtain greater homogeneity in the sizes of the population, demographic circumstances, and accuracy of demographic statistics [25]. We did not include sub-regional pooled estimates because these subregions include countries that are not participating in COSI, and therefore a sub-regional estimate would not provide an accurate assessment of the situation in that geographical area.

For each variable, we calculated the frequency of consumption according to country and sex. We tested for differences between sex in the distribution of the responses using the Rao–Scott chi-square test, a design-adjusted version of the Pearson’s χ^2^ test. We applied post-stratification weights to adjust for the sampling design, oversampling, and nonresponse proportions in order to infer results from the sample of the population. These were available and applied for all countries, with the one exception of Lithuania, where an unweighted analysis was carried out. All analyses took account of the complex survey nature of the data (i.e., multiple stages, clustering, and stratification). Pooled estimates were calculated, including only one target age group per country in order to balance the contribution of each country to the pooled estimates and to limit the differences in children’s age as much as possible. An adjusting factor was applied to the post-stratification weights to take account of differences in the population sizes of the countries involved. The adjusting factor was based on the number of children belonging to the targeted age group according to Eurostat figures or national official statistics for 2016.

A *p*-value of 0.05 was used to define statistical significance. All statistical analyses were performed in the statistical software package Stata version 15·1 (StataCorp LLC, College Station, TX, USA).

## 3. Results

A total of 132,489 children from 23 countries were included in the analysis. The number of participants per country varied widely, from 306 children in San Marino to 43,696 in Italy (Table 1). Most of the children (75.2%) were seven years of age and 51.3% were boys.

The pooled estimates indicated that most children (78.8%) consumed breakfast every day, but around 2.3% consumed breakfast “never or less than once a week” and 8.6% consumed breakfast only on “some days” (one to three days a week). The pooled estimates indicated that 42.5% of children consumed fresh fruit “every day” and 7.5% “never consumed it or consumed it less than once a week”. Around a quarter (22.6%) of all children consumed vegetables “every day”, and 14.0% consumed it “never or less than once a week.” The pooled estimates indicated that 5.2% of children consumed savory snacks “every day”, but 57.9% consumed savory snacks “never or less than once a week”. Around one in ten children (10.3%) consumed sweets “every day” and a third (32.8%) consumed sweets “never or less than once a week”. Around one in ten (9.4%) children consumed soft drinks every day.

### 3.1. Consumption of Breakfast

The percentage of children who consumed breakfast every day ranged from 48.9% in Kazakhstan to 96.4% in Portugal (Figure 1). Between-country and between-region differences in breakfast consumption were not tested for significance, although there were visible between countries and no clear patterns according to region. There were no significant differences in breakfast consumption between boys and girls (Appendix A).

### 3.2. Consumption of Fresh Fruit

The frequency of consuming fresh fruit everyday ranged widely between the regions. The consumption of fresh fruit every day was highest in the Southern European countries, with 80.8% in San Marino, 72.6% in Italy, and 63.1% in Portugal (Figure 1, Appendix A). Meanwhile, the daily fresh fruit consumption was low in the Central Asian countries—Kyrgyzstan 18.1%, Kazakhstan 33.3%, and Tajikistan 33.5%—with an exception of Turkmenistan with 70.1%. The same trend was visible for differences in the proportion of children who consumed fresh fruit “never or less than once a week”, ranging from 3.0% in San Marino, 3.3% in Portugal, and 2.2% in Montenegro to 21.4% in Tajikistan and 22.6% in Kyrgyzstan.

There were significant differences in fresh fruit consumption between sexes, with girls more likely to eat fruit on a daily basis compared to boys (Appendix A).

### 3.3. Consumption of Vegetables

Daily vegetable consumption ranged from 9.1% in Spain to 68.1% in Turkmenistan and 74.3% in San Marino (Figure 2). The percentage of children who consumed vegetables “never or less than once a week” was higher in Western Asian countries, with 20.4% in Turkey and 17.1% in Georgia compared with 1.3% in Czechia and 1.4% in Turkmenistan. There were significant between-sex differences, with boys tending to eat vegetables less frequently than girls (Appendix A).

### 3.4. Consumption of Savory Snacks (Like Potato Crisps, Corn Chips, Popcorn, or Peanuts)

We observed large differences between countries and regions of Europe in the frequency of consuming savory snacks like potato crisps, corn chips, popcorn, or peanuts. Low values for the daily consumption of savory snacks were observed in the Northern European countries, where Denmark reported 0%, Lithuania and Latvia 0.6%, and Ireland 1.5% (Figure 2). In contrast, in the Southern European and Asian countries, daily consumption of savory snacks was more frequently reported. Albania reported a percentage of 21.5% of children who consumed savory snacks every day, as well as Tajikistan with 11.3% and Montenegro and Turkmenistan with 9.0% (Appendix A). In Malta, only 7.7% of the children never or less than once week, while in the Russian Federation, 90.8% of children consumed these foods “never or less than once a week”. Similar results were seen in Lithuania (83.1% consumed savory snacks “never or less than once a week”) and in Latvia (80.2% never or less than once a week). There were no significant differences in the consumption of savory snacks between boys and girls (Appendix A).

### 3.5. Consumption of Sweets (Like Candy Bars or Chocolate)

Daily consumption of sweet snacks like candy bars or chocolate ranged from 0.4% in Denmark to 21.1% in Turkmenistan and 22.8% in Bulgaria (Figure 2). The percentage of children who never or less than once a week consumed sweet snacks ranged from 3.9% in Malta to 56.7% in Spain and 67.8% in Portugal. There were no clear regional trends in the distribution of daily sweet snack consumption and there were no significant sex differences (Appendix A).

### 3.6. Consumption of Soft Drinks

The frequency of consuming soft drinks every day was lowest in Northern European countries, with a value of 0.4% in Ireland, 2.0% in Lithuania, and 2.1% in Denmark (Figure 2). In comparison, in the Central Asian countries of Tajikistan (32.8%) and Turkmenistan (25.8%), daily soft drink consumption was relatively high among some children. There was a lower percentage of children who never or less than once a week consumed soft drinks in the Central Asian countries (33.8% in Tajikistan, 34.1% in Turkmenistan, 40.5% in Kyrgysztan, 50.3% in Kazakhstan) compared to the Northern European countries (88.0% in Ireland, 72.0% in Lithuania, 62.0% in Latvia, 53.8% in Denmark). We observed no significant differences between boys and girls (Appendix A).

## 4. Discussion

The purpose of this study was to provide a snapshot that updates the general picture of the dietary habits of European children. Our data present a largely confirmatory picture of the current understanding, with some bright spots in terms of dietary habits but also many areas of opportunity.

The bright spots include high levels of breakfast consumption, with around 80% of children consuming breakfast every day. Daily breakfast consumption was the lowest in Kazakhstan, with fewer than half of the children consuming it every day, whilst almost all the children consumed it every day in Portugal and Denmark. The pooled results align with findings from another systematic review of 286,804 children and adolescents (2 to 18 years) living in 33 countries, which found that the prevalence of skipping breakfast ranged from 10–30%, with an increasing trend in adolescents, especially girls [26]. However, there is also evidence to suggest that breakfast consumption may decrease as children get older. A recent report from the Health Behaviour in School-Aged Children (HBSC) survey of findings from 227,441 young people aged 11, 13, and 15 years living in 45 countries/regions found that more than four out of 10 adolescents do not eat breakfast every school day, and that girls across all ages tend to skip breakfast and eat fewer meals with their family than boys [27].

The areas of opportunity for improving children’s diets are related to increasing the consumption of fruits and vegetables. We found that only 42.5% of children consumed fruit and 22.6% consumed vegetables on a daily basis, but there were wide between-country differences. Daily fruit consumption ranged from 80.8% in San Marino to only 18.1% in Kyrgyzstan and 19.2% in Lithuania. Three-quarters (74.3%) of the children in San Marino consumed fresh vegetables every day compared to 9.1% of children in Spain, 11.9% in Turkey, 14.1% in Lithuania, and 14.4% in Georgia. Data on fruit and vegetable consumption trends from 33 countries participating in the HBSC surveys from 2002, 2006, and 2010 indicate that many adolescents do not consume fruit and vegetables on a daily basis, but there was an increase in daily fruit and vegetable consumption between 2002 and 2010 in the majority of countries [28]. Even so, findings from the latest HBSC report (2017/2018) found that almost two in three adolescents do not eat enough nutrient-rich foods, such as fruits and vegetables [27]. A recent review on dietary patterns found that among adolescents, the average fruit and vegetable consumption is below recommended levels in almost all populations [29].

This study also highlights the need for continued efforts to discourage the consumption of foods that are high in salt, sugar, and fat, and low in nutritional value. The pooled estimates related to the frequency of consuming savory snacks, sweets, and soft drinks suggest that 5.2%, 10.3%, and 9.4% of children consume these foods daily, respectively, but there was a wide between-country variation. The percentage of children consuming savory snacks every day ranged from 0% in Denmark and 0.1% in the Russian Federation to 21.5% in Albania. Daily consumption of sweet snacks ranged from 0.4% in Denmark to 21.1% in Turkmenistan and 22.8% in Bulgaria. Consumption of daily soft drinks ranged from 0.4% in Ireland to 32.8% in Tajikistan. This aligns with similar findings from the latest results from the HBSC, which also indicates that one in four adolescents eat sweets and one in six consume sugary drinks at least once a day [27]. Few available data have been found on savory snack consumption or sodium intake among children, but for the majority of the included populations, levels were far above the recommended five grams per day [29].

Based on the data from 23 countries, this study found wide between-country differences in children’s healthy and unhealthy eating habits, but few clear patterns according to region. These differences were likely due to a complex range of factors. Eating patterns and food preferences in childhood are shaped by individual, interpersonal, and environmental factors, including a child’s family, cultural background, social environment, socioeconomic status, and school environment [30]. Children today are increasingly exposed to environments where energy-dense and nutrient-poor foods are promoted [31] and readily available, which can make eating a healthy diet challenging. Other factors to consider include cultural differences, differences in school food environments, differences in home food environments, differences in family traditions and mealtimes, differences in the level of adherence to national dietary guidelines, price differences (which may affect the affordability and accessibility of healthy or unhealthy foods and differences in the availability of fruit or vegetables), and more.

These data lend further support to existing calls for urgent action to improve child nutrition. Schools may improve nutrition by following quality standards for school meals and providing students with access to healthy foods and beverages (such as fresh fruits, vegetables, and fresh water), and nutrition education [32]. Examples of successful initiatives include the European Union’s School Fruit and Vegetable Scheme [33] and the WHO’s Nutrition-Friendly Schools Initiative [34].

Another possible action to improve nutrition is through fiscal incentives or subsidies to promote better nutrition, both for encouraging the consumption of healthy foods (such as fruit or vegetables, or discouraging the consumption of unhealthy foods, such as sugary drinks). The United States Department of Agriculture provides reimbursement to states that operate nonprofit breakfast programs in schools [35]. Taxation on sugar-sweetened beverages has been shown to be effective in the reduction of sugar consumption [36,37]. Another potential intervention is the reformulation of processed foods [38], which has shown promise for the reduction of both sugar [39] and salt [40,41].

Current food and beverage marketing practices predominantly promote low-nutrition foods and beverages, and have a direct effect on children’s nutrition knowledge, preferences, purchase behavior, consumption patterns, and diet-related health [42]. Governments should restrict the marketing of unhealthy foods to children, particularly in the digital world, where advertising may be especially persuasive [43]. Implementation of the WHO recommendations on the marketing of foods and non-alcoholic beverages to children is one indicator in the Global Monitoring Framework [44].

A comprehensive approach involving action at many levels is required to improve children’s diets [45]. In addition to improving food environments (within schools, at home, and in other places where children gather), action is needed to engage parents and other adults who care for children [46]. Parents or caretakers often play a key role in ensuring the availability of nutritious foods, not only at home but also at school (in instances when children bring a packed lunch to school). Parents or caregivers can also help to ensure that children consume appropriate portion sizes, and there may be opportunities for governments to provide better guidance and support for parents and caregivers [47]. Front-of-pack labeling can help provide parents with information to support healthier eating choices and food purchases [48], although a recent Cochrane review suggests that more research may be needed regarding the effects that food labeling may have on consumer choices [49]. Nutritious diets may be more expensive, and this may contribute to socioeconomic disparities in health [50]; therefore, efforts must be made to ensure access to healthy and affordable foods, especially for vulnerable groups. Results from COSI indicate that the prevalence of obesity and severe obesity among children in Europe was common among children whose mothers had a lower level of education [51].

This study had several limitations. First, this study used a dietary questionnaire that has not been validated and which did not collect information on the portion sizes of foods. Future work is needed to validate this questionnaire and identify possible methods of assessing portion sizes of various foods that are consumed per day in order to identify the prevalence of children meeting certain nutrition recommendations (such as consuming five portions of fruits and vegetables per day). Further work is needed to validate questions about “sweets” and “savory snacks” where there may be cultural variation in the ways that these categories are understood. Another limitation is that the dietary indicators used in this paper are not comprehensive. For example, we reported here on “sweet treats, such as candy bars or chocolate” but there are other sweet foods, such as “biscuits, cakes, or doughnuts” (Appendix A) that were assessed in a separate question but not reported in this paper.

This study was limited by a cross-sectional study design and a reliance on parental reports of children’s diet behaviors, which may have limited accuracy [52,53]. Other limitations include possible social desirability bias or non-response bias. We do not know how the responders in this study varied from non-responders, but previous research indicates that healthier individuals with a higher socioeconomic status are more likely to respond to health surveys [54] or dietary surveys [55], and this may result in an overestimation of the prevalence of healthy behaviors. Another limitation is that this study did not account for seasonal differences in the availability of fruit and vegetables; responses to this questionnaire were collected during the autumn, winter, or spring months when fruit and vegetable availability may have been lower than during the summer months, particularly in Central Asia.

One of the main strengths of this study was that it collected data from a large sample of children across a diverse range of countries using nationally-representative sampling methods and following a common protocol.

This project is an ongoing one and will be updated in future years. It is a valuable dataset that represents the collaboration of many committed experts and provides important visibility into the habits of European children. Accurate data on children’s weight status, eating habits, and other energy-balance-related behaviors provide a vital underpinning for government actions to implement and evaluate effective and appropriate strategies to combat under-nutrition and obesity. Investment in high-quality surveillance is essential to ensure more children can benefit from good nutrition and improved health during childhood and onward through the life course.

## Figures and Tables

**Figure 1 nutrients-12-02481-f001:**
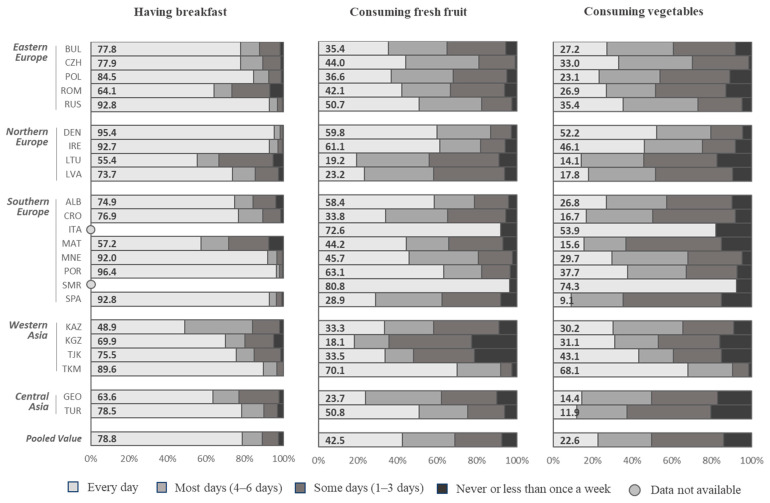
Frequency of consuming breakfast, fresh fruit, and vegetables among boys and girls by country ^a^. COSI/WHO Europe round 4 (2015–2017). ^a^ Pooled values were estimated, including the following age groups/countries: 7-year-olds from Bulgaria, Czechia, Denmark, Kyrgyzstan, Georgia, Ireland, Latvia, Lithuania, Malta, Montenegro, Portugal, Spain, Tajikistan, Turkey, and Turkmenistan; 8-year-olds from Albania, Croatia, Poland, and Romania; and 9-year-olds from Kazakhstan.

**Figure 2 nutrients-12-02481-f002:**
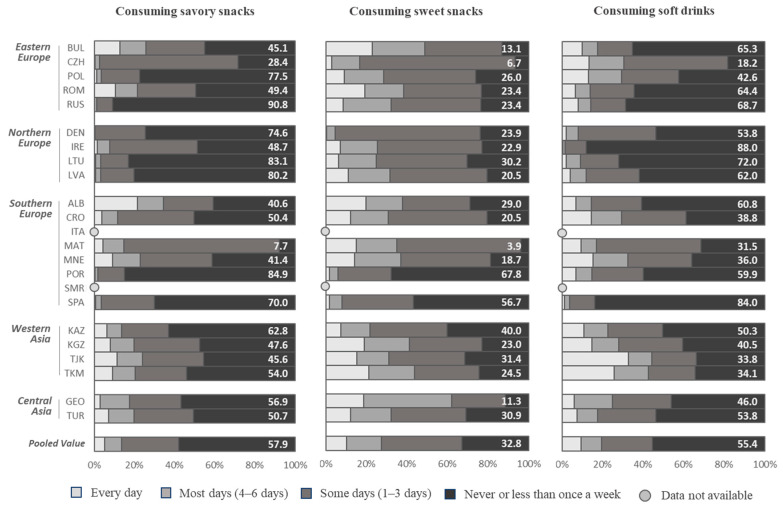
Frequency of consuming savory snacks, sweet snacks, and soft drinks among boys and girls by country ^a^. COSI/WHO Europe round 4 (2015–2017). ^a^ Pooled values were estimated, including the following age groups/countries: 7-year-olds from Bulgaria, Czechia, Denmark, Kyrgyzstan, Georgia, Ireland, Latvia, Lithuania, Malta, Montenegro, Portugal, Spain, Tajikistan, Turkey, and Turkmenistan; 8-year-olds from Albania, Croatia, Poland, and Romania; and 9-year-olds from Kazakhstan.

**Table 1 nutrients-12-02481-t001:** Number of children invited to participate in COSI/WHO Europe Round 4 (2015–2017), the number of children included in the analysis, and the percentage of children participating by sex, age, and country.

Country ^a^	Children Invited to Participate ^b^		Children Included in the Analysis ^c^			Percentage of Children Participating by Sex and Age (%) ^d^				
	Total Number	Proportion Whose Family Form Was Filled in (%)	Boys (*n*)	Girls (*n*)	Total (*n*)	Boys (%)	6-Year-Olds	7-Year-Olds	8-Year-Olds	9-Year-Olds
Albania	7113	36.2	1315	1212	2527	52.5	0.1	24.2	52.0	23.7
Bulgaria	4090	83.1	1702	1698	3400	51.5	0	100.0	0	0
Croatia ^e^	7220	76.0	1318	1333	2651	51.1	0	0	100.0	0
Czechia	n.a.	n.a.	670	736	1406	50.7	49.5	50.5	0	0
Denmark	3202	29.9	511	446	957	52.7	27.4	70.2	2.4	0
Georgia	4143	78.4	1667	1579	3246	51.2	1.6	85.1	13.0	0.3
Ireland	2704	32.4	438	436	874	52.6	38.2	60.2	1.6	0
Italy	50,902	95.2	22,425	21,271	43,696	51.5	0	0.5	66.3	33.1
Kazakhstan	6026	82.3	2149	2162	4311	50.6	0	0.4	51.0	48.6
Kyrgyzstan	8773	86.6	3798	3769	7567	50.7	10.5	43.5	39.5	6.5
Lithuania	5527	69.8	1930	1882	3812	50.6	0.4	66.4	33.0	0.2
Latvia	8143	71.5	2752	2955	5707	48.2	7.9	43.8	9.1	39.3
Malta	4329	73.4	1589	1590	3179	50.0	0.1	69.7	30.1	0.1
Montenegro	4094	66.8	1441	1295	2736	52.8	31.2	48.4	20.1	0.2
Poland	3828	76.9	1451	1494	2945	50.2	0	0	100.0	0
Portugal	7475	85.6	3167	3224	6391	50.7	25.2	49.0	24.0	1.7
Romania	9094	73.6	3312	3298	6610	49.1	0.4	28.4	47.5	23.8
Russian Federation	3900	52.6	1006	1046	2052	50.2	18.8	72.8	8.3	0.1
San Marino	329	93.6	138	168	306	45.1	0	0	64.7	35.3
Spain	14,908	70.1	5290	5163	10,453	50.9	25.4	25.2	24.9	24.6
Tajikistan	3502	93.5	1623	1647	3270	51.6	7.7	90.8	1.4	0.2
Turkmenistan	4085	95.3	1944	1947	3891	49.9	0	79.9	20.1	0
Turkey	14,164	81.7	5335	5167	10,502	50.9	11.4	82.3	6.0	0.3
Total	198,683	79.5	66,971	65,518	132,489	51.3	0	75.2	18.2	6.6

n.a.—not available. ^a^ Figures refer to primary school children from Albania (ALB), Bulgaria (BUL), Croatia (CRO), Czechia (CZH), Denmark (DEN), Georgia (GEO), Ireland (IRL), Italy (ITA), Kazakhstan (KAZ), Kyrgyzstan (KGZ), Lithuania (LTU), Latvia (LVA), Malta (MAT), Montenegro (MNE), Poland (POL), Portugal (POR), Romania (ROM), Moscow city (RUS), San Marino (SMR), Spain (SPA), Tajikistan (TJK), Turkmenistan (TKM), and Turkey (TUR). ^b^ Total figures were calculated including only countries with available information about the number of children invited to participate in the surveillance. ^c^ All children with complete information on sex, whose age was between six and nine years old and with information on eating habits from the family form. ^d^ Pooled values were estimated, including the following age groups/countries: 7-year-olds from Bulgaria, Czechia, Denmark, Kyrgyzstan, Georgia, Ireland, Latvia, Lithuania, Malta, Montenegro, Portugal, Spain, Tajikistan, Turkey, and Turkmenistan; 8-year-olds from Albania, Croatia, Poland, and Romania; and 9-year-olds from Kazakhstan. The figures were estimated by applying post-stratification weights. ^e^ For Croatia, only data on 8-year-olds were available for comparison at the European level. The proportion of children whose parents or caregivers filled in the family form was calculated in the whole sample (not only for 8-year-olds).

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
