# Peer review of "A Snapshot of European Children’s Eating Habits: Results from the Fourth Round of the WHO European Childhood Obesity Surveillance Initiative (COSI)"

_nutrients, 2020, doi:10.3390/nu12082481_

Round 1

Reviewer 1 Report

Williams et al have undertaken a study addressing a research topic with special interest to public health, that of portraying some eating behaviors of children, with the strength of a great geographical coverage. While the manuscript is generally well-written, I would recommend a more focused approach regarding some points and a refinement of some others, as suggested below.

Abstract

Methodology should be added.

Results about the total scoring would also worth mentioning.

Introduction

I would suggest Introduction was more focused on the subject of the manuscript. For example, does malnutrition really falls into the general research hypothesis of the study? Do the questions delivered to participants address malnutrition? Moreover, discussing about the environmental factors that influence dietary behavior does not seem to be in relevance with what the study measured. I find that Introduction should avoid a rather general background and focus on describing the background of the aspects the manuscript addresses.

Materials and Methods

Please clarify to which time period the questions on eating habits referred to. Was it last month, last semester?

Table 1. I would not see a need to put superscripts on score 1, as I understand that giving one point means a less healthy habit, depending on the usual frequency of each food consumption.

Lines 159-60. This grouping creates somehow a confusion. Is it really a grouping, meaning that the authors then analysed data by having two categories, or do you just describe the scoring system approach? By the way, wouldn’t be interesting to analyse data by using frequency of each behavior as a binary variable, by grouping the four frequencies into two?

Results

3.7 Eating habits score. What was the mean score, eventually?

Even though the authors clearly state that discussing obesity status is beyond the scope of this paper, it is reasonable to ask for some results to be presented regarding weight status data (which are obviously self-reported)? No need to have a thorough analysis, but the reader would inevitably question him/herself whether accumulation of non-healthy habits correlates with weight status (given the limitations the author do point out in the Discussion).

Discussion

I would propose a restructure and refinement of this section, beginning with a summary of the results (general), and then describing differences by country (more specific).

Line 354. Use of past tense wouldn’t be more appropriate?

I would also propose all limitations to be developed towards the end of the section. The limitation about fruit portions (lines 382-390) is in fact a limitation that applies to the tool used and refers to all food groups studied. More limitations should also be mentioned, such as the cross-sectional design, the self-reported data (while parental involvement attenuates this), the concept of the scoring system (as it indicates an equal importance of some frequencies with no gradual change). Please, elaborate on these.

Strengths of the study should also be included in the Discussion.

Line 454. As the authors mention non-response bias, did the non-responders actually differ from the responders, and to which parameters?

Several actions to be taken to improve dietary habits are described throughout the section, but these seem to include almost every level of action, from macroenvironment to the individual. I would welcome a more focused to the subject of the study proposal, for better cohesion. As funding comes from governmental and policy agents, I feel proposals regarding this level of action are more relevant to this article.

By the way, “classes devoted to health eating” (l. 445), what do you mean? Is it about knowledge or cooking skills? A reference would be welcome.

Tables & Figures

I find explaining the abbreviations of the countries at all captions would be helpful for the reader.

Please, check the captions, there are some “copy-paste” typos (e.g. vegetables, savoury snacks).

I am sceptic about the number of the tables. Do the authors thing it would it be an option to include some, perhaps present analysis by two categories, and then present all detailed results as a kind of an Appendix?

Reviewer 2 Report

Reviewer’s comments: A Snapshot of European Children’s Eating Habits: Results from the fourth round of the WHO European Childhood Obesity Surveillance Initiative (COSI)

Major points:

  1. My biggest concern is with the reliability and validity of the questionnaire used by parents to report the frequency of their child’s intake. As noted in the discussion section, these questions and the methods used to create variables representing frequency of consumption have not been validated. These data were collected as part of the COSI study which reports the prevalence of obesity in these countries. At the very least the association between the eating habits score and child weight or BMI should be evaluated; even using a sub-sample for these analyses would provide some evidence of the validity of the questionnaire.
  2. I am also concerned about the very limited examples that are used for some of the eating behaviors and worry that parents may not be reporting on food choices that reflect the category being asked about. Specifically, the question on sweet consumption includes only candy bars or chocolate as examples. Would parents also consider other sweet treats such as cookies, cakes and pastries when responding to this question?
  3. In general, more information on the survey development is needed in this paper. How did the authors of the survey address country and regional differences in the examples that they used for the eating behaviors? For example, did parents from all the countries involved find the examples used for savory snacks (potato crisps, corn chips, popcorn or peanuts) relevant to their cultural patterns?  Are there regional or cultural-specific foods that should have been used as examples? How was the survey developed to address these issues?
  4. The justification for the Eating Habits Score needs to be expanded. For breakfast, fruit and vegetable consumption ‘every day’ is the only response option that scores a zero while for the other eating behaviors more leniency is offered. What happens to the eating score if response options of ‘every day’ and ‘most days’ are scored as zero?
  5. It may be helpful to look at comparisons by region (Eastern Europe, Northern Europe, Southern Europe, Central Asia and Western Asia). Currently, differences are only examined by sex. The sample size for each country is sufficient to l look for differences by country but with 23 countries involved comparisons between countries may provide unwieldy results that are difficult to interpret. Examining regional differences may provide useful insights into cultural or environmental differences by region.
  6. The discussion is not very helpful and could have been written without presenting much of the data in the tables. If data are combined by region would clearer patterns of differences emerge?

Minor points:

  1. There are a lot of dense tables. Is there some way to reduce or reformat the data presented?
  2. It appears that two different criteria for identifying significance are used in Table 9. Footnote a and b use p<0.05 but Footnote C uses p<0.0001.
